# ACAP1 Deficiency Predicts Inferior Immunotherapy Response in Solid Tumors

**DOI:** 10.3390/cancers14235951

**Published:** 2022-12-01

**Authors:** Qiyi Yi, Youguang Pu, Fengmei Chao, Po Bian, Lei Lv

**Affiliations:** 1School of Basic Medical Sciences, Anhui Medical University, 81 Meishan Road, Hefei 230032, China; 2Department of Cancer Epigenetics Program, Anhui Cancer Hospital, The First Affiliated Hospital of USTC, Division of Life Sciences and Medicine, University of Science and Technology of China, Hefei 230001, China

**Keywords:** ACAP1, tumor-infiltrating lymphocytes (TILs), immune checkpoint blockade therapy (ICT), cancer immunotherapy, CD8+ T cells

## Abstract

**Simple Summary:**

ACAP1 plays a key role in endocytic recycling, which is essential for the normal function of lymphocytes. This study aimed to assess the expression and function of ACAP1 in lymphocytes. In this study, we revealed that ACAP1 is expressed primarily in lymphocytes and is necessary for the normal function of lymphocytes. Its expression is regulated by promoter DNA methylation and the transcription factor SPI1. ACAP1 levels positively correlate with the infiltrating levels of tumor-infiltrating lymphocytes (TILs) across a broad range of solid cancer types. Moreover, ACAP1 deficiency is associated with poor prognosis and immunotherapeutic response in multiple cancer types treated with immunotherapy. Thus, this study demonstrates that ACAP1 is a novel lymphocyte marker. We propose a widely applicable indicator to predict response to immunotherapy that may guide cancer patient stratification for appropriate therapy.

**Abstract:**

Background: ACAP1 plays a key role in endocytic recycling, which is essential for the normal function of lymphocytes. However, the expression and function of ACAP1 in lymphocytes have rarely been studied. Methods: Large-scale genomic data, including multiple bulk RNA-sequencing datasets, single-cell sequencing datasets, and immunotherapy cohorts, were exploited to comprehensively characterize ACAP1 expression, regulation, and function. Gene set enrichment analysis (GSEA) was used to uncover the pathways associated with ACAP1 expression. Eight algorithms, including TIMER, CIBERSORT, CIBERSORT-ABS, QUANTISEQ, xCELL, MCPCOUNTER, EPIC, and TIDE, were applied to estimate the infiltrating level of immune cells. Western blotting, qPCR, and ChIP-PCR were used to validate the findings from bioinformatic analyses. A T-cell co-culture killing assay was used to investigate the function of ACAP1 in lymphocytes. Results: ACAP1 was highly expressed in immune-related tissues and cells and minimally in other tissues. Moreover, single-cell sequencing analysis in tumor samples revealed that ACAP1 is expressed primarily in tumor-infiltrating lymphocytes (TILs), including T, B, and NK cells. ACAP1 expression is negatively regulated by promoter DNA methylation, with its promoter hypo-methylated in immune cells but hyper-methylated in other cells. Furthermore, SPI1 binds to the ACAP1 promoter and positively regulates its expression in immune cells. ACAP1 levels positively correlate with the infiltrating levels of TILs, especially CD8+ T cells, across a broad range of solid cancer types. ACAP1 deficiency is associated with poor prognosis and immunotherapeutic response in multiple cancer types treated with checkpoint blockade therapy (ICT). Functionally, the depletion of ACAP1 by RNA interference significantly impairs the T cell-mediated killing of tumor cells. Conclusions: Our study demonstrates that ACAP1 is essential for the normal function of TILs, and its deficiency indicates an immunologically “cold” status of tumors that are resistant to ICT.

## 1. Introduction

Cancer is a significant public health burden with an ever-increasing incidence rate worldwide. ICT has revolutionized cancer treatment, resulting in an outstanding clinical benefit for certain patients, especially for advanced-stage cancers. It is currently the first-line treatment for advanced melanoma and lung cancer. However, most patients have limited or no response or clinical benefits from ICT [1]. Furthermore, it might trigger severe immune-related adverse events in some patients [2]. Therefore, excavating a biomarker to predict the response to ICT is a desperate need. As the ICT re-activates TILs to recognize and attack cancer cells, the presence of TILs, such as T, B, and NK cells, in tumors is the fundamental determinant of the ICT response. Furthermore, CD8+ T cells are the most prevalent and powerful effectors in destroying neoplastic cells. Theoretically and practically, tumors with few or no TILs will not respond to ICT. Hence, it can only be utilized to treat tumors with sufficient levels of TILs. TILs as predictive and prognostic biomarkers have been under intense examination. Increasing numbers of studies suggest that the abundance of TILs, especially the T, B, and NK cells, is associated with immunotherapy response [3]. For example, higher density TILs, especially CD8+ T cells, indicated a clinical response and better prognosis in ICT-treated melanoma, renal cell cancer (RCC), and metastatic colorectal cancer [4,5,6]. Furthermore, cancer patients with high levels of TILs are prone to better prognosis and survival. For example, high densities of CD3+ and CD8+ T cells predicted a low risk of recurrence in colon cancer patients [7]. Therefore, identifying a marker gene of lymphocytes is helpful in predicting the response to ICT and patients’ prognosis.

ACAP1 (ArfGAP with Coiled-Coil, Ankyrin Repeat And PH Domains 1), also named CENTB1 (Centaurin beta1), negatively regulates the function of ARF6. It also interacts with Akt to form a protein complex, acting as an adaptor for the endocytic recycling of intracellular cargos such as transferrin receptor, cellubrevin, integrin β1, and Glut4 (glucose transporter type 4) [8,9,10,11]. It also binds to NOD1/NOD2 and then inhibits NF-κB activation in response to the stimulation of bacterial components in intestinal epithelial cells [12]. In addition, ACAP1 expression could be stimulated by pro-inflammatory cytokines and was significantly upregulated in ulcerative colitis, especially in inflammatory infiltrates [13]. A recent study showed that ACAP1 expression correlated with the infiltrating levels of immune cells in ovarian cancer, possibly by regulating a variety of immune-related pathways [14]. These studies suggest ACAP1 is highly involved in responding to physiological and pathological stimuli, such as bacteria and inflammation. However, the integrated genomic and transcriptomic analyses of ACAP1, and its role in immune response, are lacking to date.

This work systematically explores the genetic alterations, expression patterns, prognostic significance, and transcriptional regulation of ACAP1 across various cancer types. Furthermore, we investigate the relationship of ACAP1 expression with TILs and the ICT response of patients. We identify ACAP1 as a lymphocyte-specific gene and show that ACAP1 is indispensable for the normal function of lymphocytes. Its deficiency ubiquitously correlates with the “cold” immune phenotype and indicates resistance to immunotherapy in solid tumors. Based on these results, we identified a pan-cancer predictor of TILs and immunotherapy response, which could help clinicians optimize treatment regimens for cancer patients.

## 2. Materials and Methods

### 2.1. ACAP1 Expression in Tissues and Cell Lines

Gene expression for ACAP1 in normal human tissues was queried through the GTEx (Genotype-Tissue Expression) portal, a large-scale database containing 54 types of normal tissues [15]. Gene expression for ACAP1 in cancer cell lines was obtained from the “Expression 21Q4 Public” file using the DepMap portal, including 27 types and 1377 cell lines [16]. Gene expression for ACAP1 in human and mouse tissues/cell lines was also queried by BioGPS using the “GeneAtlas U133A, gcrma” [17] and “GeneAtlas MOE430, gcrma” [18] datasets, respectively.

### 2.2. TCGA Datasets

Clinical data, RNA sequencing (level 3 HTSeq-FPKM), DNA methylation data, and copy number variation (CNV) data across 33 tumor types were obtained from TCGA (The Cancer Genome Atlas). The clinical data analyzed included T, N, M, pathologic stage, and survival information. The TCGA cancer types and corresponding abbreviations are seen in Appendix A. TPM was calculated from RPKM and was then log_2_(x + 1) transformed. The CpGs located on the ACAP1 promoter were identified using xena-browser [19]. The promoter methylation scores of ACAP1 were calculated as the average β-value of cg13295242, cg13670306, cg11807006, and cg25671438, which are on the ACAP1 promoter. 

### 2.3. Single-Cell Sequencing Datasets

One glioblastoma dataset (‘Neftel cohort’ [20]), two prostate cancer datasets (“He 2021” [21] and “Wu 2021 (PA-P1)” [22]), one melanoma dataset (“Wu 2021 (M-P1)” [22]), and three breast cancer datasets (“Wu 2021 (BC-P1)” [22], “Wu 2021 (BC-P2)” [22], and “Wu 2021 (BC-P3)” [22]) were obtained from previously published articles and visualized through the Broad Single Cell Portal with corresponding cluster annotations. The other two melanoma datasets, including GSE72096 [23] and GSE115978 [24], were obtained from the GEO platform. 

### 2.4. Immunotherapy Datasets

Eight immunotherapy-related datasets were acquired for subsequent analyses of the relationship of ACAP1 expression with immunotherapy response and the prognosis of immunotherapy-treated patients. These datasets were: (I) four melanoma cohorts, including “VanAllen 2015” [25], “Gide 2019” [26], “Snyder 2014” [27], and “Riaz 2017” [28]; (II) two lung cancer cohorts, including “Ruppin 2021” [29] and GSE126044 [30]; (III) “Miao 2018”, a renal cell carcinoma cohort [31]; and (IV) IMvigor210, a bladder cancer cohort [32]. The processed gene expression data of ACAP1 and clinical information, including immunotherapy response and prognosis, were collected from cBioPortal, GEO platform, or supplementary files of original publications. The expression data of ACAP1 deposited with the publication was used without further processing. Only samples with both expression data and response/prognosis information were used. The determinations of “response” were acquired from original publications or designated as CR (complete response) or PR (partial response), whereas “non-response” was assumed for PD (progressive disease) or SD (stable disease). In Table 1, fifteen datasets were used, including “VanAllen 2015” [25], “Riaz 2017” [28], “Nathanson 2017” [33], “Liu 2019” [34], “Lauss 2017” [35], “Hugo 2016” [36], “Gide 2019” [26], “Ruppin 2021” [29], “Kim 2018” [37], “Miao 2018” [31], “McDermott 2018” [5], “Braun 2020” [38], “Zhao 2019” [39], “Mariathasan 2018” [32], and “Uppaluri 2020” [40].

### 2.5. Other Datasets

The protein levels of ACAP1 in tumors and corresponding paracancerous tissues were compared using CPTAC (Clinical Proteomic Tumor Analysis Consortium) datasets through UALCAN. [41] All available protein datasets were analyzed, including breast cancer, ccRCC, colon cancer, GBM, LIHC, HNSC, LUAD, OV, PAAD, and UCEC.

To confirm the association of ACAP1 levels with prognosis in cancer patients from TCGA, four datasets, including ICGC-LIRI-JP [42], GSE68465 [43], GSE22153 [44], and CGGA325 [45], were obtained for analysis. Only samples with both RNA-sequencing and survival data were analyzed. ICGC-LIRI-JP is a hepatocellular carcinoma dataset obtained from ICGC (International Cancer Genome Consortium). GSE68465 and GSE22153, downloaded from the GEO platform, are lung adenocarcinoma and melanoma datasets, respectively. The CGGA325 dataset was downloaded from CGGA (Chinese Glioma Genome Atlas), and GBM (glioblastoma multiforme) samples in CGGA325 were selected for analysis. 

### 2.6. ChIP-Sequencing Analysis and JASPAR Analysis

The ChIP-sequencing data was analyzed using Cistrome and visualized with the UCSC Genome Browser. The SPI1 datasets used were: GSM1681425 [46], GSM1703900 [47], and GSM1480737 [48]. The H3K4me3 datasets used were: GSM1519189 [49], GSM945267 [50], GSM1003561 [51], GSM1574256 [52], GSM2533937 [51], GSM1420153 [53], GSM2187240 [54], GSM2534207 [51], GSM3120522 [55], and GSM2309428 [56].

The SPI1-binding sites surrounding the ACAP1 promoter region were predicted using JASPAR (https://jaspar.genereg.net/; accessed on 1 July 2022). All SPI1 motifs in the JASPAR database were used for binding enrichment analysis using a significance threshold of 0.90. 

### 2.7. Calculation of Immune Cell Infiltration

The “Immune-Gene” module of TIMER2.0-Cistrome was utilized to explore Spearman’s correlation between ACAP1 mRNA levels and immune infiltrates across all TCGA tumors, except LAML [57]. Eight algorithms, including TIMER [58], CIBERSORT [59], CIBERSORT-ABS [59], QUANTISEQ [60], xCELL [61], MCPCOUNTER [62], EPIC [63], and TIDE [64], were used for estimation in this analysis. The results were displayed as heatmaps.

### 2.8. Survival Analysis and Gene Set Enrichment Analysis (GSEA) 

Survival analyses were ascertained using Kaplan–Meier methodology and analyzed by the log-rank test with the R (version 3.6, R Foundation, Vienna, Austria) packages “survminer” (https://CRAN.R-project.org/package=survminer/, accessed on 10 September 2021) and “survival” (https://CRAN.R-project.org/package=survival/, accessed on 10 September 2021). The patients were dichotomized into high and low groups based on an optimal cutoff value, which was determined with the “survminer” package.

Spearman’s correlation analyses of ACAP1 with all other genes were performed across TCGA tumors. The calculated correlation coefficient in each tumor type was used for GSEA analysis using GSEA software (version 4.1.0) with the following parameters: database = c5.go.bp.v7.4.symbols.gmt, number of permutations = 1000, enrichment statistic = weighted, max size = 500, min size = 3. The genes involved in the main associated pathways were obtained from the MsigDB (https://www.gsea-msigdb.org/gsea/msigdb/index.jsp, accessed on 28 September 2021) and summarized in Appendix A.

### 2.9. Cell Culture and Lentivirus Transfection

Jurkat, TALL-104, OS-RC-2, Caki-1, RT112/84, J82, KU1919, BFTC905, UM-UC-3, 5637, SK-HEP-1, Huh-7, A549, H1299, Calu-3, H1975, KYSE-30, KYSE-140, KYSE-150, KYSE-410, and KYSE-450 were cultured in RPMI1640 + 10% FBS (GIBCO). Recombinant human IL-2 (ProteinTech, Wuhan, China, #HZ-1015) was added to the TALL-104 culture media with a final concentration of 100 units/mL. A-498 was cultured in MEM + 10% FBS. HCT116, HT-29, SW480, and SW620 were cultured in DMEM + 10% FBS. Cells were obtained from Shanghai Cell Bank (Shanghai, China) or ATCC.

For SPI1 overexpression, the coding sequence of the SPI1 genes was amplified by PCR and sub-cloned into pSIN-3×Flag for lentivirus production. For ACAP1 knockdown, the sequences of two shRNAs targeting human ACAP1 were obtained from The RNAi Consortium (MISSION^®^ TRC shRNA library, Sigma-Aldrich, St Louis, MO, USA) and sub-cloned into the pLKO.1 vector. shRNA1: CCGGGCAGGAGATGAGACGTATCTTGGATCCAAGATACGTCTCATCTCCTGCTTTTTG; shRNA2: CCGGTCACGCTAAATACGTGGAGAAGGATCCTTCTCCACGTATTTAGCGTGATTTTTG.

The pSIN-3×Flag-SPI1 or ACAP1 shRNAs viruses were generated by the transfection of the constructs together with pMD2.G and psPAX2 into HEK293T cells using Attractene Transfection Reagent (Qiagen, Hilden, Germany, #301007) to package virus particles. Then, the cells were transduced with recombinant lentivirus suspension (+8 μg/mL polybrene) and selected with 1 μg/mL puromycin to obtain stable cell lines.

### 2.10. Cell Treatment

For the demethylation treatment, SK-HEP-1 and Huh-7 cells were treated with 30 μM 5-aza (Sigma-Aldrich, #A3656) or DMSO for five days. Then, total RNA was harvested, reverse-transcribed, and analyzed by qRT-PCR.

For the hypoxia treatment, Jurkat cells were treated with vehicle or hypoxia mimetic CoCl_2_ (100 and 200 μM) for 8 h and 24 h. Cells were then harvested for subsequent Western blotting.

### 2.11. ChIP-PCR

A ChIP assay was carried out in the Flag-SPI1-overexpressing Jurkat cells using a ChIP kit (Beyotime, Beijing, China, #P2078) as the manufacturer’s instructions. The Flag antibody (Sigma, St Louis, MO, USA, #F7425) was used to enrich the DNA fragments containing the putative SPI1 binding site of the ACAP1 promoter, and the normal IgG (CST, #2729) was used as a negative control. Primers used for ChIP-PCR were: 5′-ACTGCCTGGAAGTGTGGGGT-3′ and 5′-ATGCAGGTGGAGGCACTTTCT-3′.

### 2.12. qRT–PCR

Total RNA was harvested, reverse-transcribed, and analyzed by qRT-PCR as described before [65]. The primer sequences used were as follows. ACAP1: 5′-CCTGACTCAGAAAGGCGGTTCT-3′ and 5′-CATCAAGGCGAGCCTGACTGAA -3′; GAPDH: 5′-GGAGCGAGATCCCTCCAAAAT-3′ and 5′-GGCTGTTGTCATACTTCTCATGG-3′. Experiments were performed five times.

### 2.13. Western Blotting

The assay was performed as previously described [65]. The primary antibodies: ACAP1 (ProteinTech, #66596-1-Ig), SPI1 (ProteinTech, #66618-2-Ig), HIF1α (ProteinTech, #20960-1-AP), and GAPDH (ProteinTech, #60004-1-Ig). The secondary antibodies: HRP-conjugated Goat Anti-Mouse IgG(H+L) (ProteinTech, #SA00001-1) and HRP-conjugated Goat Anti-Rabbit IgG(H+L) (ProteinTech, #SA00001-2). The original WB images are provided as Appendix A).

### 2.14. T-Cell Co-Culture Killing Assay

The assay was carried out as described [66]. Briefly, the lung cancer cells A549 were co-cultured with control or ACAP1-shRNA1/2 TALL-104 cells at a ratio of 1:2 for 24 h. Then, the medium was aspirated off, and the cells were washed with PBS. Green fluorescent dye CMFDA (5 µM) (Aladdin, Shanghai, China, #C131098) and red fluorescent dye propidium iodide (10 µM) (Aladdin, #P113815) were used to stain and detect live and dead cells using fluorescence microscopy (Olympus, Tokyo, Japan, IX73). 

### 2.15. Statistical Analysis

Wilcoxon rank-sum test or two-sided Student’s *t*-tests were used to compare the difference between two groups, and Wilcoxon signed-rank test or paired *t*-test was used for paired comparisons. Spearman correlation was used for all correlation analyses. Log-rank test was used for survival analyses. All analyses were conducted in GraphPad (version 9.3.0) or R (version 3.6.0). *p*-values < 0.05 were considered significant. 

## 3. Results

### 3.1. ACAP1 Is a Marker Gene for Lymphocytes

We first examined the expression distribution of ACAP1 in normal human tissues using the GTEx portal. ACAP1 exhibited a tissue-specific expression pattern across different tissues. Its expression was enriched in immune-related tissues and cells, including whole blood, spleen, and lymphocytes. It showed a moderate expression level in the small intestine and lung but minimal expression in other tissues (Figure 1A). Expression analysis in human tissues/cells through interrogating BioGPS also revealed the high expression of ACAP1 in immune-related tissues and cells, especially CD8+ T cells, CD4+ T cells, B lymphoblasts, and NK cells (Figure 1B), with similar expression profiles in mouse tissues and cells (Appendix A). Consistently, analysis of ACAP1 expression in cancer cell lines from the “Cancer Cell Line Encyclopedia” showed that its expression was high in hematopoietic and lymphoid cell lines but barely expressed in other cell lines (Figure 1C). Western blotting of 26 kinds of cell lines also indicated that the ACAP1 protein expression was high in Jurkat cells, an immortalized T lymphocyte cell line, while under-detectable in other cell lines, including kidney cancer (OS-RC-2, Caki-1, A-498), bladder cancer (RT112/84, J82, KU1919, BFTC905, UM-UC-3, 5637), liver cancer (SK-HEP-1, Huh-7, HepG2), colon cancer (HCT116, HT-29, SW480, SW620), lung cancer (A549, H1299, Calu-3, H1975) and esophageal cancer (KYSE-30, KYSE-140, KYSE-150, KYSE-410, KYSE-450) (Figure 1D).

To better understand the expression pattern of ACAP1, we extended the analysis to the single-cell level. Analysis of two melanoma single-cell sequencing cohorts, including GSE72056 and GSE115978, revealed that ACAP1 was highly expressed in lymphocyte cells, including T, B, and NK cells but was barely expressed in malignant cells and other types of immune cells (Figure 2A,B). The expression of ACAP1 was undetectable in more than 95% of malignant cells by single-cell RNA sequencing but in only ~40% of T, B, and NK cells (Appendix A). Then, seven single-cell sequencing datasets, including a glioblastoma dataset (Neftel 2019), two prostate cancer datasets (“He 2021” and “Wu 2021 (PC-P1)”), a cutaneous melanoma dataset (“Wu 2021 (M-P1)”), and three breast cancer datasets (“Wu 2021 (BC-P1)”, “Wu 2021 (BC-P2)” and “Wu 2021 (BC-P3)”) were also examined. Different types of cells were clustered and visualized using previously established tSNE (t-distributed stochastic neighbor embedding) or UMAP (uniform manifold approximation projection) parameters. Consistently, ACAP1 expression was significantly enriched in T, B, and/or NK cells (Figure 2 and Appendix A). 

Altogether, these collective results suggest that ACAP1 is predominantly expressed in lymphocytes, including T, B, and NK cells, and may be a marker gene for TILs.

### 3.2. Pan-Cancer Expression Analysis of ACAP1

We then profiled the expression of ACAP1 in tumor samples and corresponding normal samples across 33 TCGA cancer types by integrating the sequencing data from the TCGA and GTEx datasets. It revealed that ACAP1 was significantly decreased in most cancer types (17 of 33) compared to normal samples, including BLCA, BRCA, COAD, DLBC, GBM, KICH, LGG, LIHC, LUAD, LUSC, PRAD, READ, TGCT, THCA, THYM, UCEC, and UCS. Conversely, ACAP1 expression was significantly increased in six cancer types: CHOL, HNSC, KIRC, KIRP, LAML, and PAAD, compared to normal samples (Figure 3A). Pairwise comparisons between tumor and tumor-adjacent samples in the TCGA cohorts showed that ACAP1 expression was significantly lower in eight cancer types (BLCA, BRCA, COAD, KICH, LUSC, PRAD, THCA, and UCEC) but higher in four cancer types (CHOL, HNSC, KIRC, and STAD) (Figure 3B). It is noted that most of these results from paired comparisons were consistent with the results from non-paired comparisons, except for STAD. Subsequently, we explored the protein level of ACAP1 using datasets from the CPTAC. The protein level of ACAP1 was significantly lower than normal tissues in three cancer types, including breast cancer, colon cancer, and LIHC (Figure 3C) but significantly higher than corresponding normal tissues in four cancer types, including ccRCC, HNSC, PAAD, and UCEC (Figure 3C). 

To determine whether ACAP1 expression reflects cancer progression, we assessed ACAP1 expression in different T, N, M, and pathologic stages. It revealed that the low expression of ACAP1 tended to be accompanied by advanced stages in most cancer types (Appendix A). For example, ACAP1 mRNA expression in LUAD was gradually downregulated according to T, N, M, and pathologic stage progression. 

Altogether, the ACAP1 expression in cancers was lower than in normal or paracancerous tissues in the majority of cancer types, although the differential expression between them varies across cancer types. Moreover, low ACAP1 levels were often associated with advanced stages.

### 3.3. Prognostic Value of ACAP1 Expression

We next evaluated the prognostic value of ACAP1 mRNA expression for cancer patients by Kaplan–Meier analysis of OS (overall survival), DSS (disease-specific survival), and PFI (progression-free interval). 

The OS analyses across TCGA datasets indicated that low ACAP1 levels correlated with shorter survival in patients with BLCA, BRCA, CESC, HNSC, LIHC, LUAD, OV, PAAD, PCPG, SARC, SKCM, STAD, THYM, UCEC, and UCS (Figure 4A–O) but more prolonged survival in patients with DLBC, GBM, KIRC, KIRP, LAML, and UVM (Figure 4P–U). Interestingly, a trend towards better OS was observed in hematological malignancies (DLBC and LAML) in patients expressing low ACAP1 expression. To validate the OS result in TCGA, several non-TCGA datasets were used for analysis. Low ACAP1 levels predicted shorter OS in the LIHC datasets ICGC-LIRI-JP, GSE68465 (LUAD dataset), and GSE22153 (SKCM dataset) but longer OS in the CGGA325 (GBM dataset), which was consistent with TCGA findings. 

The DSS analysis in TCGA demonstrated that patients with low ACAP1 levels possessed significantly worse DSS than those with high ACAP1 levels in ACC, BLCA, CESC, DLBC, HNSC, LIHC, LUAD, PAAD, SARC, SKCM, UCEC, and UCS but better DSS in GBM and KIRC (Appendix A). The PFI analysis indicated that patients with low ACAP1 levels had significantly shorter PFI in ACC, BLCA, BRCA, CESC, CHOL, DLBC, HNSC, LIHC, SKCM, PAAD, UCEC, and UCS but longer PFI in KIRC, GBM, and PRAD than those with high ACAP1 levels (Appendix A).

Overall, these results indicate that, despite the prognostic significance of ACAP1 expression varied by tumor type, low ACAP1 expression was associated with poor clinical outcomes in the majority of tumor types, especially solid tumors.

### 3.4. Transcriptional Regulation of ACAP1

Next, we attempted to explore the underlying mechanisms that regulate ACAP1 expression. DNA methylation is a prominent regulatory mechanism for gene expression. There are four CpG sites, including cg13295242, cg13670306, cg11807006, and cg25671438, in the ACAP1 promoter (Figure 5A). We found that cg25671438 was hypomethylated in three immune-related cancer types (DLBC, LAML, and THYM) with a high expression level of ACAP1, while it was hypermethylated in other cancer types that possessed a low ACAP1 expression level (Figure 3A and Figure 5B). Analysis of 1028 kinds of cell lines from the GSE68379 dataset showed that cg25671438 was hypomethylated in most blood cells but hypermethylated in other cell lines (Figure 5C). Furthermore, the average methylation level (β-value) of these four CpG sites was also significantly lower in blood cells than in other cells (Figure 5D). Spearman correlation analysis revealed that ACAP1 expression was significantly negatively correlated with the average methylation level of the ACAP1 promoter in 29 cancer types of TCGA. There was also an inverse correlation between ACAP1 expression and the β value of these CpG sites in a variety of tumor types, especially cg25671438 site (Figure 5A,E). Analysis from the GSE68379 dataset showed that three CpG sites (cg13670306, cg11807006, and cg25671438) were hypermethylated in Huh-7 and SK-HEP-1, two liver cancer cells (Figure 5F). After treatment with 5-aza, a DNA demethylating agent, ACAP1 mRNA levels increased in Huh-7 and SK-HEP-1 cells (Figure 5G). These results suggest that DNA methylation negatively regulates ACAP1 expression at the transcription level. 

CNV (copy number variation) often positively regulates gene expression. ACAP1 deep deletion and amplification, both of which are CNV, were observed in most cancer types (25 of 32) (Appendix A). Spearman correlation analyses in TCGA cohorts indicated that ACAP1 expression was significantly positively correlated with its copy number in 23 types of cancer (Figure 5A,E), suggesting that it was at least partially regulated by CNV in these tumors.

Transcription factors also play a key role in the regulation of gene expression. As a result, we investigated whether SPI1 (PU.1), a major transcription factor involved in immune cell activity, modulates ACAP1 expression. Spearman correlation analysis indicated a strong positive correlation between ACAP1 and SPI1 expression in the TCGA pan-cancer dataset (Figure 5A) and 28 types of cancer from TCGA (Figure 6A). Moreover, analysis from GTEx also showed that SPI1 expression positively correlated with ACAP1 expression in EVB-transformed lymphocytes and whole blood significantly (Figure 6B,C). Thus, we postulated that SPI1 transcriptionally regulates the ACAP1 gene. Analysis from three different SPI1 ChIP-seq datasets, including GSM1681425 in macrophage [46], GSM1703900 in B lymphocyte [47], and GSM1480737 in lymphoma [48], demonstrated that SPI1 binds to the ACAP1 promoter (Figure 6D). Moreover, analysis from ten ChIP-seq datasets of H3K4me3, an active promoter mark, indicated that the ACAP1 promoter is active in B and T cells but inactive in the lung, pancreatic ductal, cervix, colon, prostate, esophagus, brain, and breast tissues (Figure 6D), which was accompanied by the hypermethylation of the ACAP1 promoter in these tissues (Figure 5). Furthermore, analysis through JASPAR showed that SPI1 was predicted to bind to the ACAP1 promoter with two motifs (MA0080.1 and MA0080.2) (Figure 6D,E). Then, ChIP-PCR analysis was carried out on Flag-SPI1 overexpressing Jurkat cells by using specific antibodies against Flag, showing the enrichment of SPI1 at the ACAP1 promoter, which validated the ChIP-seq results (Figure 6F). Moreover, SPI1 overexpression significantly increased the ACAP1 levels in Jurkat cells (Figure 6G). Interestingly, hypoxia, an important tumor microenvironmental stimulus, increased the expression of both SPI1 and ACAP1 in Jurkat cells (Figure 6H).

Taken together, our results suggest that ACAP1 expression is positively modulated by CNV and SPI1 but negatively regulated by promoter methylation. 

### 3.5. ACAP1 Expression Is Positively Associated with TILs and Is Essential for the Cytotoxicity of Lymphocytes

To identify the specific pathways associated with ACAP1 expression, we initially performed pathway enrichment analysis using GSEA for each TCGA cancer type based on ACAP1 expression in TCGA cohorts. The analyses suggested that ACAP1 expression was strongly associated with multiple immune-related “Biological Processes”, such as “B cell-mediated immunity”, and “humoral immune response mediated by circulating immunoglobulin”, in almost all tumor types, except two immune-related tumor types (LAML and THYM) (Appendix A). 

As we have demonstrated that ACAP1 is a lymphocyte marker, combined with these GSEA results, we hypothesized that ACAP1 could be a suitable, sensitive, and broadly applicable indicator of the infiltrating level of TILs. We then systematically analyzed the relationship between ACAP1 expression and the immune infiltrates across diverse cancer types in TCGA, except LAML, via TIMER2, which estimates the abundances of immune infiltrates using eight immune deconvolution algorithms, including xCell, CIBERSORT, CIBERSORT-ABS, EPIC, MCPCOUNTER, QUANTISER, TIMER, and TIDE. All of these immune deconvolution analyses showed that ACAP1 was significantly positively correlated with the level of CD8+ and CD4+ T cells in most cancer types, except DLBC (Figure 7). In addition, xCell analysis showed that ACAP1 expression was positively correlated with levels of “CD8+ central memory T cells” and “CD8+ effector memory T cells” in the majority of solid tumors. It also positively correlated with levels of B cells, NK cells, dendritic cells, monocytes, and macrophages in solid tumors (Figure 7 and Appendix A). Notably, ACAP1 expression was positively correlated with the infiltrating level of immunostimulatory cells, such as “NK cell activated” and pro-inflammatory “M1 macrophage”, but negatively correlated with the level of immunosuppressive cells, including “NK cell resting” and “myeloid-derived suppressor cell (MDSC)” (Figure 7 and Appendix A). 

Given the high ACAP1 levels in lymphocytes, we postulated that ACAP1 might play an essential role in the immune response of these cells. TALL-104 is a lymphocyte cell line that expresses markers characteristic of both NK cells and cytotoxic T-lymphocytes [66,67]. The cytotoxicity of TALL-104 cells after ACAP1 downregulation was assessed by a co-culture experiment (Figure 8A). Interleukin-2 activated pre-treated TALL-104 was co-cultured with the lung cancer cells A549 at a 2:1 effector-to-target (E:T) cell ratio for 24 h. Then cell-mediated killing was evaluated via the quantification of dead and live tumor cells. The A549 cells were found to be more resistant to ACAP1-knockdown TALL-104 cells with a much lower dead/live cell ratio compared with the control TALL-104 (Figure 8B,C). The result indicated that loss of ACAP1 impaired the cytotoxicity of lymphocytes to tumor cells. 

Moreover, we found that ACAP1 expression was strongly positively correlated with cytotoxicity score (CYT) and the expression of GZMA and PRF1, which encode two key cytolytic effectors (granzyme A and perforin 1), in most cancer types (Appendix A). It was also positively correlated with the expression of MHC genes, which could present tumor-associated antigens to CD8+ T cells, in the majority of cancer types of TCGA (Appendix A). 

Thus, the analyses of immune infiltrates from different methods consistently pinpointed that ACAP1 expression was significantly positively correlated with the level of TILs, especially CD8+ T cells, in the majority of cancer types. Meanwhile, ACAP1 is essential for the cytotoxicity of lymphocytes. 

### 3.6. ACAP1 Level Correlates with Immunotherapy Efficacy and Predicts Prognosis in Cancer Patients Treated with ICT

As ACAP1 expression was strongly correlated with immune cytotoxicity activity and the infiltrating level of CD8+ T cells in tumors at a pan-cancer level, we wondered whether ACAP1 expression could predict the tumor response to ICT and the prognosis of ICT-treated patients. Firstly, we utilized the ‘Biomarker evaluation’ module from TIDE (Tumor Immune Dysfunction and Exclusion) (http://tide.dfci.harvard.edu/setquery/, accessed on 13 September 2022) to determine the performance of ACAP1 expression in predicting the ICT response. We also compared its predictive power with existing biomarkers, including the TIDE score, the MSI score, tumor mutational burden (TMB), and CD274 (PD-L1), by calculating the AUC (area under the ROC curve). As shown in Table 1, the prediction performances of ACAP1 were better than TIDE, MSI, TMB, and CD274 in >50% of melanoma cohorts and achieved a high prediction accuracy of ICT response with an AUC > 0.7, predicting a strong likelihood of positive response to immunotherapy, in four cohorts. Notably, ACAP1 exhibited good power in predicting ICT response in lung cancer (AUC = 0.8286, “Ruppin 2021” cohort), pronouncedly superior to other biomarkers. ACAP1 exhibited a modest predictive performance and was comparable to other biomarkers in gastric cancer, ccRCC, glioblastoma, and mUC (metastatic urothelial cancer) (Table 1). However, it was not predictive in the HNSC cohorts (AUC < 0.4, “Uppaluri 2020” cohort) (Table 1).

Next, we investigated its association with ICT response and prognosis in eight ICT datasets, including four melanoma datasets (“VanAllen 2015”, “Snyder 2014”, “Gide 2019”, “Riaz 2017”, and “Lauss 2017”), two lung cancer datasets (“Ruppin 2021” and GSE126044), a kidney cancer dataset (“Miao 2018”), and an mUC dataset (IMvigor210). All sequencing samples were collected before therapy, except for a subset of patients in the “Riaz 2017” cohort. Analysis from “VanAllen 2015” and “Snyder 2014” cohorts showed that ACAP1 expression in the non-response group was lower than in the response group of melanoma patients receiving anti-CTLA-4 therapy. Furthermore, low ACAP1 expression was associated with worse OS and PFS (progression-free survival) in these patients (Figure 9A,B). Analysis from the “Gide 2019” cohort revealed that low ACAP1 levels were also associated with inferior response, OS, and PFS in melanoma patients treated with anti-PD-1 alone or in combination with anti-CTLA-4 antibodies (Figure 9C). Analysis from the “Riaz 2017” cohort showed that low ACAP1 levels were associated with worse response and OS in melanoma patients treated with anti-PD-1 antibodies regardless of whether the samples were collected prior to or during therapy (Figure 9D). Further, analysis from two lung cancer cohorts (“Ruppin 2021” for LUAD and GSE126044 for NSCLC) revealed that low ACAP1 expression was significantly associated with decreased response rate and worse OS/PFS in patients treated with anti-PD-1 antibodies (Figure 9F,G). Analysis from other ICT datasets, including an RCC dataset and an mUC dataset, showed that low ACAP1 expression was associated with inferior response, OS, and PFS in these patients, although this was not highly significant (Figure 9E,H).

Overall, these results suggest that low ACAP1 levels are associated with a decreased response rate and worse clinical outcomes in ICT-treated patients, highlighting the potential clinical significance of ACAP1 for identifying patients who will benefit from ICT across multiple cancer types, especially for melanoma and lung cancer patients.

## 4. Discussion

Previous studies of ACAP1 focused mainly on its role in the endocytic recycling of intracellular cargos. Other biological functions of ACAP1, especially in immune response, have barely been studied. In this study, we found that ACAP1 expression was highly enriched in T, B, and NK cells but at a low level in other cells, indicating that ACAP1 is a novel marker gene for lymphocytes. Its expression is highly correlated with infiltrating levels of TILs, especially CD8+ T cells, and its deficiency defines a “cold” immune microenvironment across all solid tumors. Moreover, ACAP1 is essential for the cytotoxicity of lymphocytes. Consequently, the low expression of ACAP1 indicated resistance to immunotherapy and was associated with poor outcomes in multiple cancer patients following ICT, especially in patients with melanoma or lung cancer. Thus, we excavated a biomarker to predict the response to ICT.

Pan-cancer expression analysis of ACAP1 showed that ACAP1 was significantly decreased in most cancer types but significantly increased in several cancer types, such as CHOL, HNSC, and KIRC. We also demonstrated that ACAP1 levels represent the abundance of lymphocytes in tissues. Combining the above results, we speculated that lymphocyte was excluded and decreased in most types of cancer, such as LUAD. However, the abundance of lymphocytes increased in CHOL, HNSC, and KIRC. We speculated that some factors that may attract immune cell infiltration are significantly increased. For example, HNSC is a type of cancer characterized by high TMB [68,69], which could induce a large neoantigen load that could attract immune cell infiltration [70]. 

Accumulating evidence suggests that a low level of TILs is often associated with poor cancer progression and survival [3,71]. In the current study, we found that low ACAP1 expression was associated with worse OS in 15 solid tumors, including BLCA, BRCA, CESC, HNSC, LIHC, LUAD, OV, PAAD, PCPG, SARC, SKCM, STAD, THYM, UCEC, and UCS, as well as poor DSS and shorter PFI in most of these tumors. Its low expression was associated with better OS in only six tumors, including four solid tumors (GBM, KIRC, KIRP, and UVM) and two hematological malignancies (DLBC and LAML). However, low ACAP1 expression was associated with worse DSS and PFI in DLBC. These results indicated that high ACAP1 expression is a protective factor for patients with the majority of tumors. We showed that ACAP1 is a lymphoid-lineage-specific biomarker, and a high ACAP1 level correlates with poor overall survival in LAML. This indicates that a high lymphocyte count may be an indicator of poor overall survival in LAML. In 1991, Edward et al. found that positive lymphocyte surface markers (CD2 and CD19) were associated with a more favorable prognosis in AML patients [72]. Furthermore, two other studies showed that a high ALC (absolute lymphocytic count) was associated with the prolonged overall survival of AML patients [73,74]. However, a recent study showed that a high ALC was associated with shorter remission and decreased relapse-free and overall survival in AML patients who responded to induction therapy [75]. In addition, AML patients with higher ALC showed a lower response rate to induction therapy [73]. There was a low frequency of NK cells and a high frequency of inhibitory T regulatory cells in the high ALC group [75], which may result in weaker immune responses against residual leukemia remaining after induction therapy and thus, might explain the poorer outcome of the high ALC group in AML patients after induction therapy. Another possible reason is that lymphocytes in AML are usually exhausted [76,77,78,79]. In GBM and KIRC, low ACAP1 was also associated with better OS, DSS, and PFI, suggesting that a low abundance of TILs was associated with a better prognosis in these two tumors. One possible explanation for this phenomenon may be that the TILs in GBM and KIRC are profoundly exhausted and ineffective, as has been demonstrated in GBM [80]. We noticed that ACAP1 expression is higher in KIRC than in normal tissues, and high ACAP1 expression was associated with advanced T, N, M, and pathologic stages and poor prognosis in KIRC. However, its high expression portended a better prognosis in KIRC patients treated with immunotherapy (Figure 9E). Therefore, it is worthwhile to study more deeply the changes in the status and function of TILs in KIRC. 

We compared the predictive ability of ACAP1 level with several other biomarkers for immunotherapy response. We found that its prediction performance is good in lung cancers and melanoma and is comparable to TMB, MSI, CD274, and TIDE scores in gastric cancer, ccRCC, glioblastoma, and metastatic urothelial cancer. In addition, ACAP1 expression does not highly correlate with TMB in the majority of tumors (Appendix A), such as melanoma, which has the highest TMB [81]. Furthermore, not only TIL intensity but also the status of TILs, which could be characterized by TIDE score, was important to the efficacy of immunotherapy. Therefore, the combination of ACAP1 expression, TMB, and TIDE score for immunotherapy response and survival would be a better predictive marker, which warrants further investigation. In addition, we merely analyzed the association between the transcript levels of ACAP1 and the response/outcome of immunotherapy but did not analyze the effect of ACAP1 protein expression. Further analysis of large-scale protein sequencing or immunohistochemistry of ACAP1 is needed to evaluate their relationship.

The presence of TILs, such as T, B, and NK cells, in tumors is the fundamental determinant of the ICT response, as ICT functions through re-activating TILs to recognize and attack cancer cells. Tumors with low ACAP1 levels probably do not respond to ICT. We speculate that ICT combined with TILs infusion might be superior to ICT alone for these patients. However, this should be verified by follow-up preclinical and functional studies. More work is required to optimize the clinical efficiency before it could be considered a practical treatment option.

Previous studies showed that ACAP1 acts as an adaptor for the endocytic recycling of intracellular cargos, such as transferrin receptor and integrin β1 (CD29) [8,9,10,11]. Cell surface transferrin receptor is required for B and T cell activation and regulates cell-mediated immune responses [82,83]. Integrin β1 is a representative adhesion molecule for cell–cell and cell–extracellular-matrix interactions, and it plays a crucial role in lymphocytes. For example, integrin β1 on T cells could induce FAK-mediated signaling and subsequent CD40L expression, then enhance T cell activation [84,85]. Integrin β1 also marked CD4+ and CD8+ T cells maintaining the cytotoxic phenotype [86,87]. It is required for optimal proliferation and the cytotoxicity function of NK cells [88,89]. Moreover, integrin β1 expression correlates with the expression of markers for tumor-infiltrating lymphocytes and better long-term survival of patients with melanoma, suggesting a role in anti-tumor response [90]. Therefore, we speculate that ACAP1 is also necessary for the activation and function of these cells and that the loss of ACAP1 in lymphocytes would impair their ability to kill tumor cells, leading to poor immunotherapy outcomes. In this study, we demonstrated that ACAP1 expression is required for the cytotoxicity of T cells to tumor cells. However, whether ACAP1 affects lymphocyte function by regulating the endocytic recycling of transferrin receptor, integrin β1, or other ways needs further study. In addition, it is reported that ACAP1 knockdown inhibits cell migration [9]. The migration of lymphocytes, cytotoxic CD8 T cells in particular, into a tumor is essential for its infiltration and for providing efficient defense against tumor cells, thus affecting the outcome of immunotherapy for cancer. Therefore, we speculate that lymphocytes with high expression levels of ACAP1 are more likely to infiltrate into tumors, thus killing tumor cells more effectively. Future studies are required to investigate the possibility.

## 5. Conclusions

In summary, our data demonstrate that ACAP1 is a marker of TILs, and its expression was regulated by CNV, promoter methylation, and SPI1. Its deficiency is a pan-cancer indicator of immune-cold tumors, conferring resistance to ICT and inferior outcomes following ICT. The quantitation of ACAP1 expression could help determine appropriate therapies for cancer patients, although many preclinical studies need to be done before clinical application.

## Figures and Tables

**Figure 1 cancers-14-05951-f001:**
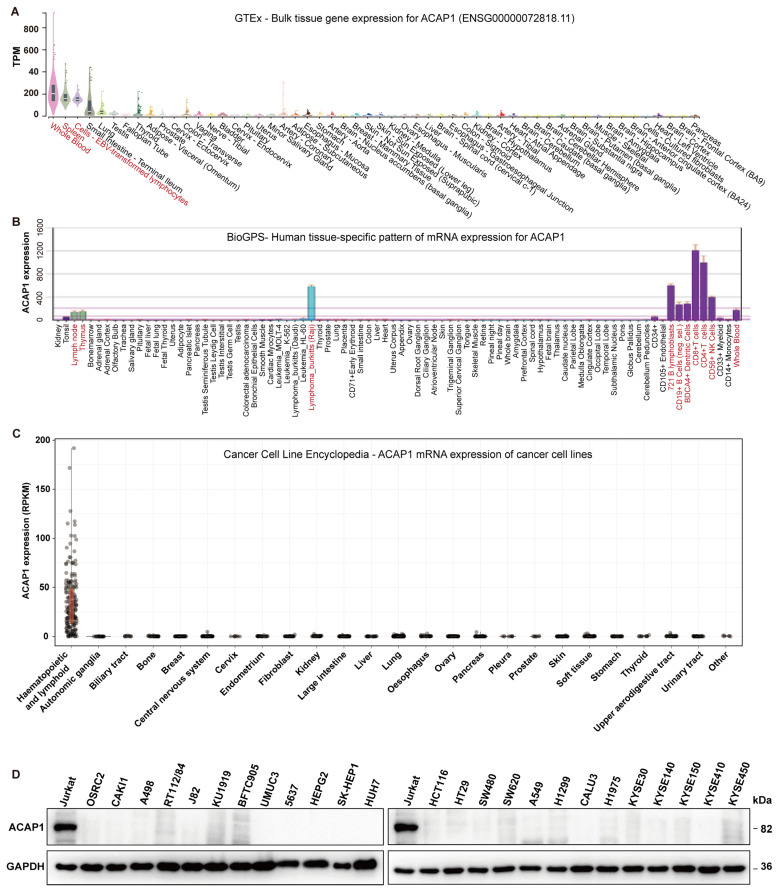
ACAP1 mRNA expression across different tissues and cell lines. (**A**) Violin plots of ACAP1 expression levels across all available tissues ordered by ACAP1 expression in the GTEx Portal. (**B**) ACAP1 expression levels in human tissues and cell lines were visualized by BioGPS. Red: tissues and cells with relatively high ACAP1 expression. (**C**) Violin plots of ACAP1 expression levels across different types of cancer cell lines in the CCLE dataset. (**D**) The protein levels of ACAP1 from indicated cell lines were determined by Western blotting.

**Figure 2 cancers-14-05951-f002:**
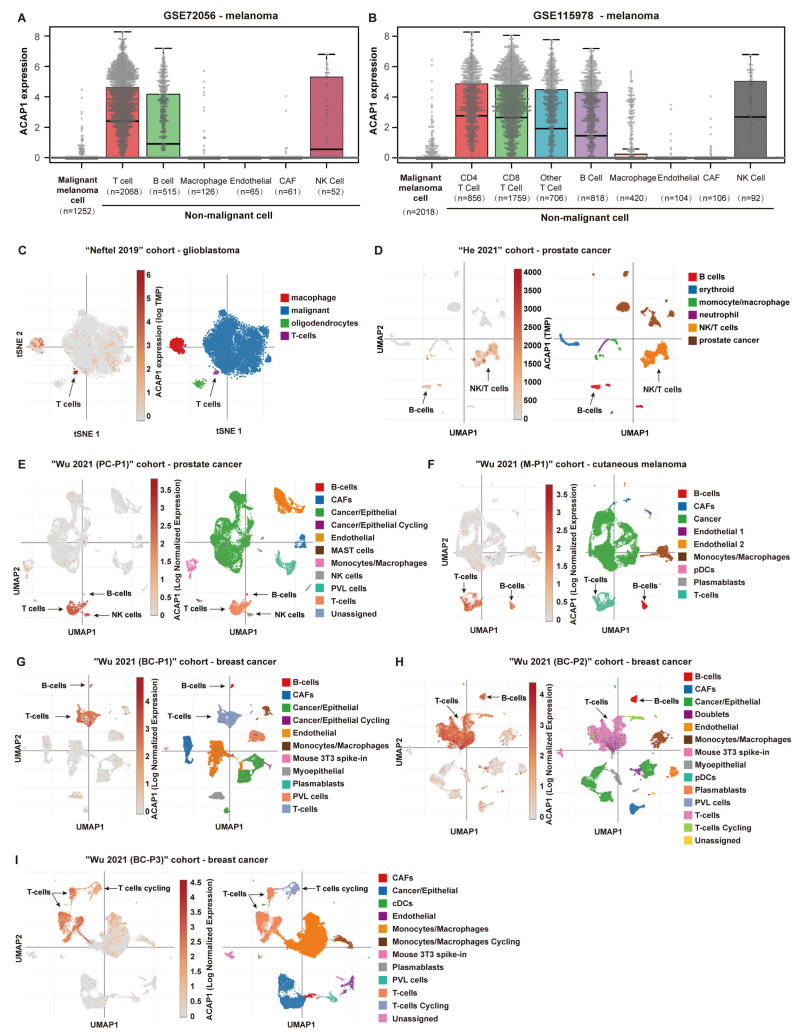
Single-cell gene expression analysis of ACAP1. (**A**,**B**) Single-cell RNA sequencing analyses of ACAP1 mRNA expression across various cell types in melanoma datasets GSE72096 and GSE115978. (**C**) Single-cell expression patterns of ACAP1 in the glioblastoma dataset “Neftel 2019” are shown with tSNE plots. (**D**,**E**) Single-cell expression patterns of ACAP1 in prostate cancer datasets, including “He 2021” and “Wu 2021 (PC-P1)”, are shown with UMAP plots. (**F**) Single-cell expression patterns of ACAP1 in the cutaneous melanoma dataset “Wu 2021 (M-P1)” are shown with UMAP plots. (**G**–**I**) Single-cell expression patterns of ACAP1 in breast cancer datasets, including “Wu 2021 (BC-P1)”, “Wu 2021 (BC-P2)”, and “Wu 2021 (BC-P3)”, are shown with UMAP plots.

**Figure 3 cancers-14-05951-f003:**
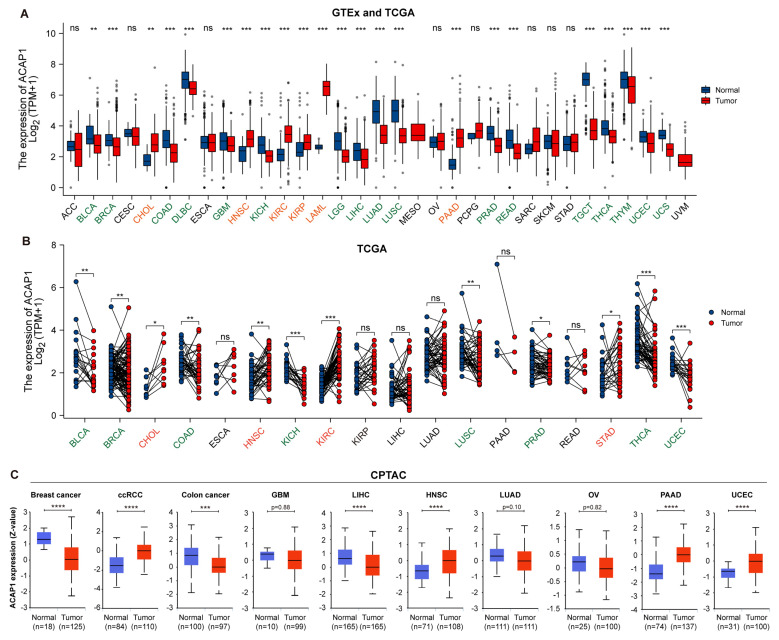
Pan-cancer analysis of ACAP1 expression in human cancer. (**A**) Comparing of ACAP1 mRNA levels in tumor vs. normal samples across TCGA cancer types by combing the TCGA and GTEx data. (**B**) Paired comparison of ACAP1 mRNA levels in tumor vs. normal samples in TCGA. Green: decreased ACAP1 expression in tumors; Red: elevated gene expression in tumors. (**C**) Comparison of ACAP1 protein levels in tumor vs. normal samples across all cancer types available in CPTAC using UALCAN webtool. **** *p* < 0.0001, *** *p* < 0.001, ** *p* < 0.01, * *p* < 0.05, ns (non-significant).

**Figure 4 cancers-14-05951-f004:**
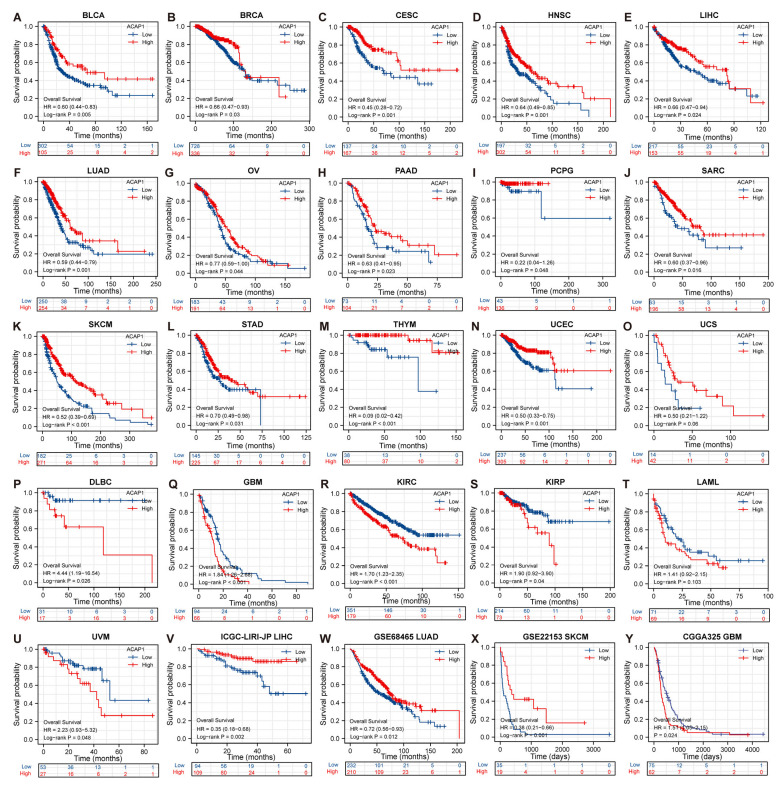
Implications of ACAP1 expression on overall survival of cancer patients across multiple cancer types. (**A**–**U**) Overall survival analyses of cancer patients stratified by ACAP1 mRNA level with the Kaplan–Meier method in TCGA datasets. (**V**–**Y**) Overall survival analyses of cancer patients stratified by ACAP1 mRNA level in ICGC-LIRI-JP(LIHC), GSE68465(LUAD), GSE22153(SKCM), and CGGA325(GBM) datasets.

**Figure 5 cancers-14-05951-f005:**
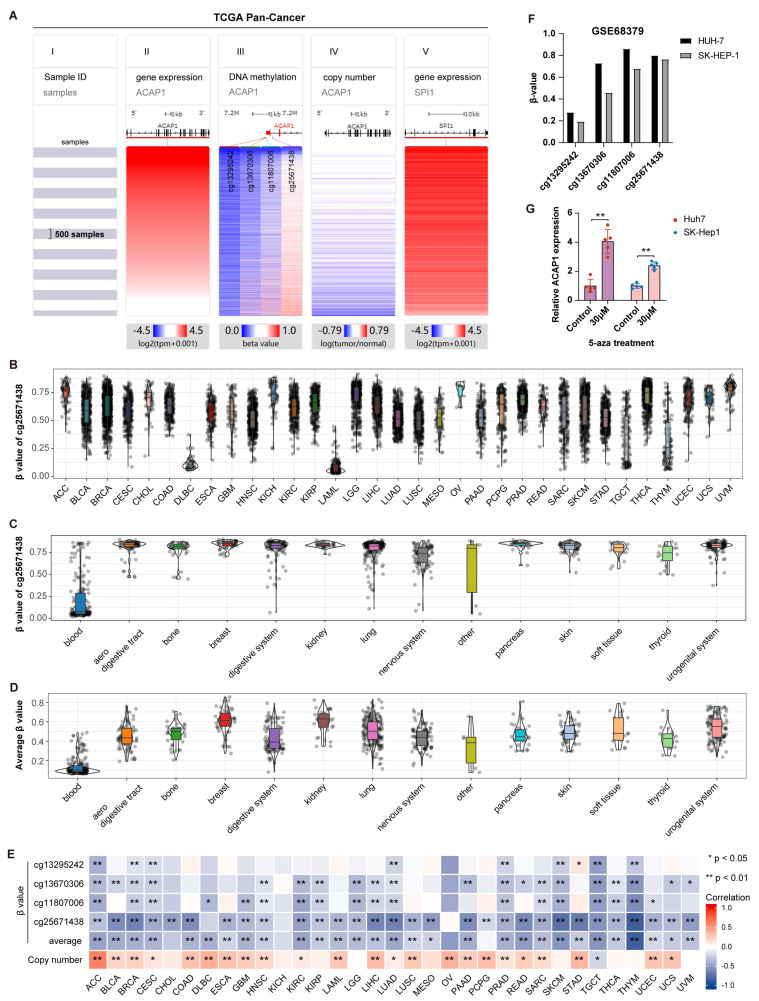
Transcriptional regulation of ACAP1. (**A**) Heatmap of TCGA samples (I), ACAP1 mRNA expression (II), β-value (methylation level) of 4 CpG sites, including cg13295242, cg13670306, cg11807006, and cg25671438, in the ACAP1 promoter region (III), copy number variation (IV), and SPI1 mRNA expression (V) in TCGA pan-cancer dataset. The samples were ordered by ACAP1 expression. Blue: low level; Red: high level. (**B**) Violin plots showing the β-value of cg25671438 in indicated cancers of TCGA. (**C**,**D**) Violin plots showing the β-value of cg25671438 and the average β-value of 4 CpG sites in indicated cancer cell lines of GSE68379. (**E**) Heatmap of the Spearman correlation coefficient of ACAP1 mRNA levels with β value of CpG sites in ACAP1 promoter and copy number across 33 cancer types in TCGA. (**F**) The β-value of 4 CpG sites in the ACAP1 promoter region of Huh-7 and SK-HEP-1 cells in GSE68379. (**G**) The impact of 5-aza on ACAP1 mRNA level in Huh-7 and SK-HEP-1 cells. ** *p* < 0.01, * *p* < 0.05.

**Figure 6 cancers-14-05951-f006:**
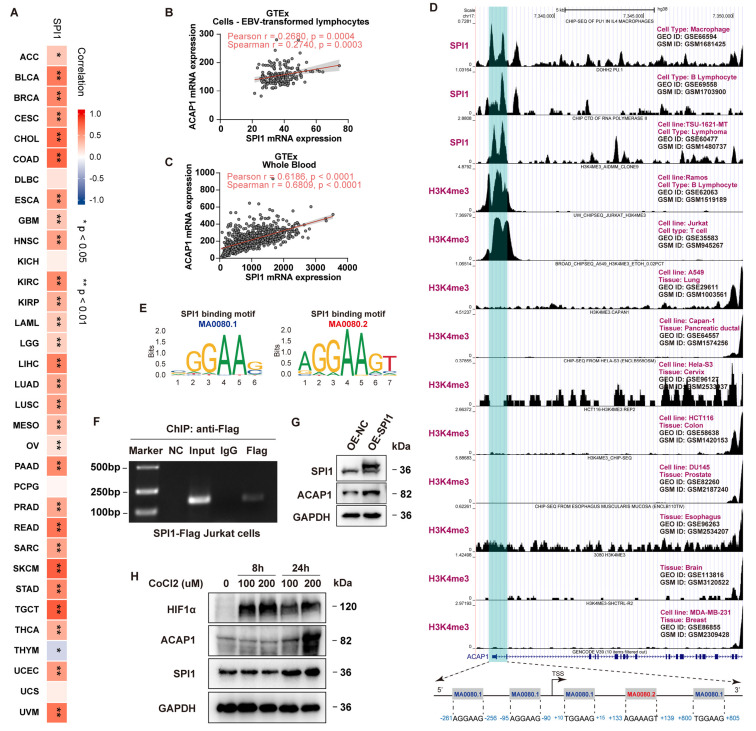
Transcriptional regulation of ACAP1 by SPI1. (**A**) Heatmap of the correlation coefficient of ACAP1 mRNA levels and SPI1 levels across 33 cancer types in TCGA. (**B**) Scatter plot displays the SPI1 and ACAP1 mRNA expression in EBV-transformed lymphocytes of GTEx dataset. (**C**) Scatter plot displays the SPI1 and ACAP1 mRNA expression in whole blood of GTEx dataset. (**D**) ChIP-sequencing peaks of SPI1 in macrophage, B lymphocyte, and lymphoma; H3K4me3 in Ramos B-lymphocytes, Jurkat T-cell, A549 lung cancer cells, Capan-1 pancreatic ductal cancer cells, Hela-S3 cervix cancer cells, HCT116 colon cancer cells, DU145 prostate cancer cells, esophagus cells, brain cells, and MDA-MB-231 breast cancer cells. The binding region of SPI1 on ACAP1 promoter is highlighted in cyan shaded box. The SPI1 binding sites on ACAP1 promoter predicted by JASPAR are shown. (**E**) SPI1 binding motif MA0080.1 and MA0080.2 from JASPAR curated motif database. (**F**) ChIP-PCR showed SPI1 binds to the promoter of ACAP1 in Jurkat cells. (**G**) Analysis of SPI1 overexpression on ACAP1 protein expression in Jurkat by Western blotting. (**H**) Effects of hypoxia-mimicking CoCl_2_ treatment on HIF1α, SPI1, and ACAP1 expression in Jurkat cells were determined by Western blotting, ** *p* < 0.01, * *p* < 0.05.

**Figure 7 cancers-14-05951-f007:**
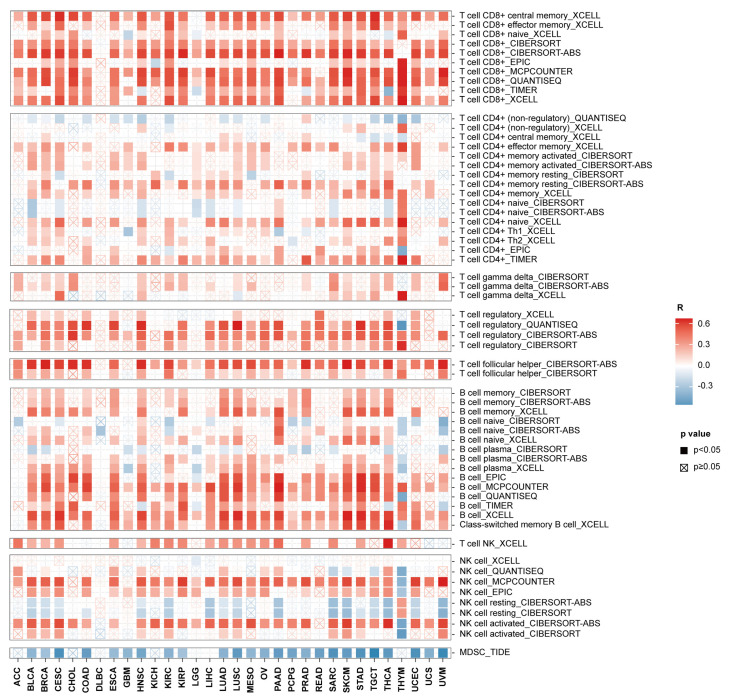
The Spearman correlations of ACAP1 expression with immune cell infiltration across 32 cancer types in TCGA.

**Figure 8 cancers-14-05951-f008:**
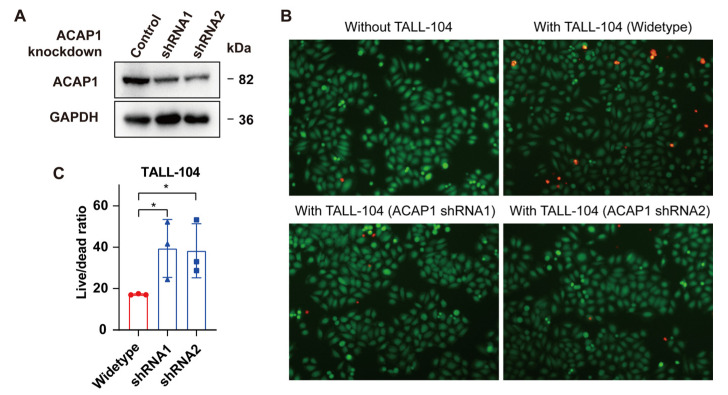
ACAP1 knockdown impairs the cytotoxicity of T cells against tumor cells. (**A**) Western blotting of lysates from TALL-104 cells infected with control or two different ACAP1-targeting shRNA lentiviruses. (**B**) Representative images of live/dead A549 cells co-cultured with different TALL-104 cells at a 1:2 cell ratio for 24 h were shown. Red-fluorescent PI (propidium iodide) was used to detect dead cells. Green-fluorescent CMFDA (5-chloromethylfluorescein diacetate) was used to detect live cells. Scale bars, 100 μm. (**C**) Three random fields were analyzed, and live/dead cell ratios were quantified. * *p* < 0.05.

**Figure 9 cancers-14-05951-f009:**
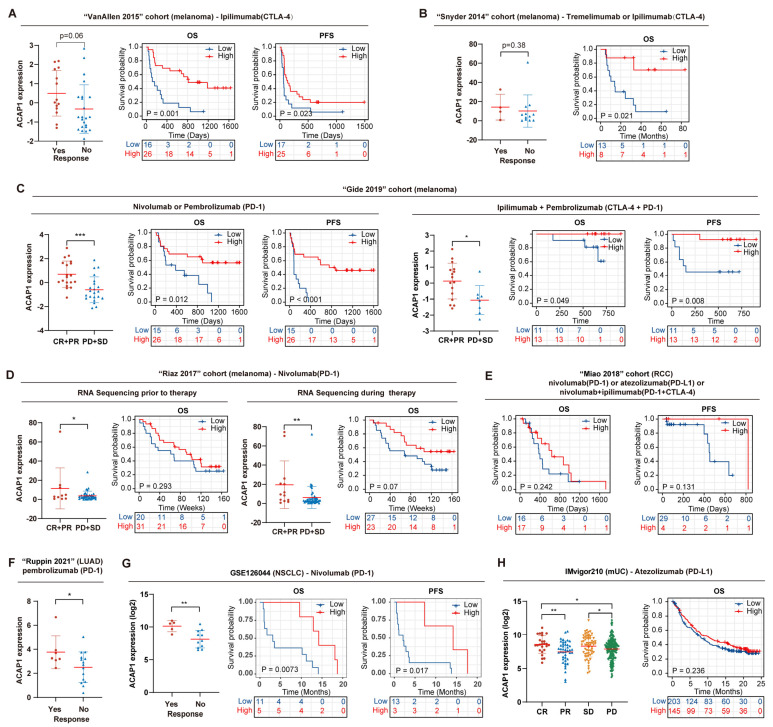
ACAP1 deficiency correlates with inferior ICT response and prognosis in multiple cancer types. (**A**) ACAP1 expression in different response groups; Kaplan–Meier OS and PFS estimates according to ACAP1 expression in “VanAllen2015” cohort, of which melanoma patients were treated with anti-CTLA-4 antibody (ipilimumab). (**B**) ACAP1 expression in different response groups; Kaplan–Meier OS estimates according to ACAP1 expression in “Snyder 2014” cohort, in which melanoma patients were treated with anti-CTLA-4 antibody (tremelimumab or ipilimumab). (**C**) ACAP1 expression in different response groups; Kaplan–Meier OS and PFS estimates according to ACAP1 expression in the “Gide 2019” cohort, in which melanoma patients were treated with anti-PD1 antibody (nivolumab or pembrolizumab) or anti-CTLA-4/PD-1 antibody (ipilimumab + pembrolizumab) (one patient with the extreme value of ACAP1 expression was excluded). (**D**) ACAP1 expression in different response groups; Kaplan–Meier OS estimates according to ACAP1 expression in both “prior to therapy” and “during therapy” groups of “Riaz 2017” cohort, in which melanoma patients were treated with anti-PD1 antibody (nivolumab). (**E**) Kaplan–Meier OS and PFS estimate according to ACAP1 expression in the “Miao 2018” cohort, in which RCC patients were treated with anti-PD-1 and/or anti-CTLA-4 antibodies (nivolumab or atezolizumab or nivolumab + ipilimumab). (**F**) ACAP1 expression in different response groups in the “Ruppin 2021” cohort, of which LUAD (lung adenocarcinoma) patients were treated with anti-PD-1 antibody (pembrolizumab). (**G**) ACAP1 expression in different response groups; Kaplan–Meier OS and PFS estimates according to ACAP1 expression in GSE126044, in which NSCLC patients were treated with anti-PD-1 antibody (nivolumab). (**H**) ACAP1 expression in different response groups; Kaplan–Meier OS estimates according to ACAP1 expression in the “IMvigor210” cohort, in which mUC patients were treated with anti-PD-L1 antibody (atezolizumab). OS: overall survival. PFS: progression-free survival. Kaplan–Meier survival curves with *p*-values derived by log-rank test were shown. *** *p* < 0.001, ** *p* < 0.01, * *p* < 0.05.

**Table 1 cancers-14-05951-t001:** TIDE biomarker evaluation of ACAP1 in response to immunotherapy across diverse cancers.

Study	Cancer Type	Treatment	Number of Pos/Neg Cases	AUC				
				ACAP1	TIDE	MSI Score	TMB	CD274
VanAllen 2015 [25]	Melanoma	CTLA4	Pos = 19, Neg = 23	0.7002	0.8032	0.7391	0.673	0.6407
Riaz 2017 [28]	Melanoma	PD1_Prog	Pos = 4, Neg = 22	0.8295	0.2273	0.6932	0.4722	0.5227
Riaz 2017 [28]	Melanoma	PD_Naive	Pos = 6, Neg = 19	0.4474	0.5965	0.4035	0.6204	0.2675
Nathanson 2017 [33]	Melanoma	CTLA4_Pre	Pos = 4, Neg = 5	0.3	0.6	0.95	N/A	0.65
Nathanson 2017 [33]	Melanoma	CTLA4_Post	Pos = 4, Neg = 11	0.75	0.25	0.5227	N/A	0.6591
Liu 2019 [34]	Melanoma	PD1_Prog	Pos = 16, Neg = 31	0.6371	0.4617	0.4456	N/A	0.5625
Liu 2019 [34]	Melanoma	PD1_Naive	Pos = 33, Neg = 41	0.5632	0.5506	0.5018	N/A	0.51
Lauss 2017 [35]	Melanoma	ACT	Pos = 10, Neg = 15	0.6867	0.54	0.4933	0.7571	0.7333
Hugo 2016 [36]	Melanoma	PD1	Pos = 14, Neg = 12	0.5179	0.7024	0.6905	0.6346	0.6012
Gide 2019 [26]	Melanoma	PD1	Pos = 19, Neg = 22	0.8158	0.6005	0.4306	N/A	0.8278
Gide 2019 [26]	Melanoma	PD1 + CTLA4	Pos = 21, Neg = 11	0.6494	0.6753	0.697	N/A	0.7879
Ruppin 2021 [29]	NSCLC	PD1	Pos = 7, Neg = 15	0.8286	0.5143	0.4571	N/A	0.6952
Kim 2018 [37]	Gastric cancer	PD1	Pos = 12, Neg = 33	0.6338	0.5985	0.75	N/A	0.8333
Miao 2018 [31]	ccRcc	PD1 or PD-L1 + CTLA4	Pos = 20, Neg = 13	0.5769	0.4808	0.2538	0.65	0.4231
McDermott 2018 [5]	ccRcc	PD-L1	Pos = 20, Neg = 61	0.6057	0.5311	0.5541	0.5357	0.6213
Braun 2020 [38]	ccRcc	PD1	Pos = 201, Neg = 94	0.449	0.4641	0.5289	0.5631	0.5621
Zhao 2019 [39]	Glioblastoma	PD1_Pre	Pos = 8, Neg = 7	0.5	0.59	0.41	N/A	0.68
Zhao 2019 [39]	Glioblastoma	PD1_Post	Pos = 6, Neg = 3	0.6667	0.6667	0.6667	N/A	0.6111
Mariathasan 2018 [32]	metastatic urothelial cancer	PD-L1	Pos = 68, Neg = 230	0.4866	0.5175	0.5551	0.7278	0.5818
Uppaluri 2020 [40]	HNSC	PD1_Pre	Pos = 8, Neg = 15	0.3667	0.4833	0.6333	N/A	0.6917
Uppaluri 2020 [40]	HNSC	PD1_Post	Pos = 9, Neg = 13	0.359	0.5385	0.453	N/A	0.7009

Note: AUC, the area under the ROC curve; Pos: positive response to immunotherapy; Neg: negative response to immunotherapy; TMB, tumor mutational burden; MSI, microsatellite instability; ACT, adoptive T-cell therapy; N/A: not available.

## Data Availability

All datasets were obtained from published literature or publicly available databases. Other data generated and the code used in this study are available from the corresponding authors upon request.

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
