# Peer review of "ACAP1 Deficiency Predicts Inferior Immunotherapy Response in Solid Tumors"

_cancers, 2022, doi:10.3390/cancers14235951_

Round 1
Reviewer 1 Report
In this manuscript, the authors have evaluated the role of ACAP1 in across various cancer types. They first discovered ACAP1 is specifically expressed in immune-related tissues, including whole blood, spleen, and lymphocytes, indicating a marker gene for lymphocytes. Next, they examined the expression level of ACAP1 across multiple data sets and concluded that ACAP1 expression were decreased in most types of cancer compared with normal tissues. Then, they evaluated the prognostic value of ACAP1 and discovered that ACAP1 expression was negatively correlated with clinical outcome. Furthermore, they explored the regulation mechanism of ACAP1 expression and found CNV and SPI1 positively regulated ACAP1 expression, while its promoter methylation negatively regulated it. Lastly, they identified the pathways involved in ACAP1 expression and found ACAP1 is essential for the cytotoxicity of lymphocytes and correlated with immunotherapy efficacy. Overall, they concluded that ACAP1 was the marker of TILs and predictor for immunotherapy outcome which could help clinicians to determine appropriate treatment for cancer patients. However, there are some remaining questions to be answered:
1, As mentioned in the article, ACAP1 expression was increased in some types of cancer and high expression level correlated with long survival rate. Could the authors briefly discussed the possible underlie reasons? Why ACAP1 shows different patterns in certain cancer types?
2, With pan-cancer expression analysis, have the authors evaluated ACAP1 expression based on the stage levels (I-IV) rather than TNM system? Does lower ACAP1 expression correlated with advanced stage (for example IV)?
3, For transcriptional regulation, have the authors evaluate the protein expression of ACAP1 after 5-aza treatment on HuH-7 and SK-HEP1 cells?
Author Response
We feel great thanks for your professional review work on our article. As you are concerned, there are several problems that need to be addressed. According to your nice suggestions, we have made some corrections to our previous draft.
Point-to-point response:
- As mentioned in the article, ACAP1 expression was increased in some types of cancer and high expression level correlated with long survival rate. Could the authors briefly discussed the possible underlie reasons? Why ACAP1 shows different patterns in certain cancer types?
Response: We think this is an excellent suggestion. As suggested, we added such discussion in the Discussion section (lines 525-557).
- With pan-cancer expression analysis, have the authors evaluated ACAP1 expression based on the stage levels (I-IV) rather than TNM system? Does lower ACAP1 expression correlated with advanced stage (for example IV)?
Response: We evaluated ACAP1 expression based on pathologic stages (Figure S7). The analysis showed that lower ACAP1 expression correlated with advanced stage in some types of tumors, such as LUAD, TGCT.
- For transcriptional regulation, have the authors evaluate the protein expression of ACAP1 after 5-aza treatment on HuH-7 and SK-HEP1 cells?
Response: Thank you for your suggestion. As suggested, we detected the change of ACAP1 protein level of HuH-7 and SK-HEP1 after 5-aza treatment through western blotting. However, we found that its protein level was still very low after treatment, which was not detected in western blotting. This may be due to the low expression level of transcription factor SPI1 in these two cells.
Reviewer 2 Report
Identification of biomarkers that could influence the outcome of immunotherapy is a highly sought, clinically relevant information. In the present study authors present substantial data from in silico models and key data from wet lab experiments to show that ACAP1 is a marker of lymphoid lineage cells present in TME and describe the important determinants of regulation of ACAP1 across tissues and cell lines.
There are very few reports describing the role of ACAP1 in anti-tumor immunity and as a potential biomarker for immunotherapy prognosis for cancers. Present study stands out in describing ACAP1 as pan cancer TME biomarker for predicting the outcome of immunotherapy. The manuscript is well written with substantial in silico data with apt in vitro data. However, the manuscript is replete with inconsistencies pertaining to references in the manuscript. Several references are missing altogether. Please see my specific comments below for all the details.
Specific comments:
It is interesting to know that ACAP1 is a lymphoid lineage specific biomarker and not a myeloid lineage as per your results described in section 3.5. However, in Section 3.3 the results describe prolonged survival in LAML which is a myeloid leukemia. Please explain why is it so?
In section 3.5 it is mentioned that ACAP1 expression is negatively correlated with M2 Macrophages. Please explain which data point signifies this in figure7.
In discussion section, it is wrong to say ACAP1 studies focused mainly on their role in endocytic recycling function. Please include a description of reports describing ACAP1 as a prognostic biomarker of tumor immunity in recent years including the report on CTNNB1 as a potential biomarker for immunotherapy prognosis in patients with hepatocellular carcinoma.
Please discuss if ACAP1`s biomarker roles are dependent on its known functions of endocytic recycling and cellular migration for immunotherapy outcomes in the discussion section.
In Fig. S11 pertaining to the results of GSEA analysis, the genes involved in different pathways is not clear, so please add a caption describing main pathways associated.
Expand ACAP1 and CENTB1 at their first mention in the manuscript.
Expand TIDE for general audience.
Please provide legend for table1 and briefly describe the performance criteria. Also, what is the significance of highlighting some rows in the table? The relevance of number of positive and negative cases is not clear, please describe this in the text.
Indicate the reference numbers for references mentioned in table 1 so it becomes easy for readers to find them.
Please insert space between author names and year in the table and elsewhere in the manuscript.
There is no reference available for following mentioned in table 1. Please revise and provide appropriate references.
Nathanson2017
Liu2019.
Lauss2017
Hugo2016
Ruppin2021
Kim2018
Braun2020
Zhao2019
Uppaluri2020
Correct the statement in the legend of figure2: Notations in green represent decreased expression in tumor whereas notations in red represent increased gene expression in tumor.
Be consistent with the notation of HuH-7 and SK-HEP-1 cells.
Provide appropriate reference of xena-browser used for identification of CpG islands on ACAP1 promoters.
Line 115, 434- reference Ruppin2021 is not included in the list of references. Reference #28 is wrongly mentioned as Ruppin 2021.
Line 114- Reference 26 is not Snyder 2014. It is absent in the reference list.
Line 292 – paracancerous is one word. Correct it.
3.5 heading should read – ACAP1 expression is positively associated with TILs and is essential for... , please correct accordingly.
Line 438 – Reference Uppaluri 2000 is missing in references.
Line 440, 441, 444 – References – Snyder 2014, Lauss 2017 are missing in reference list.
Correct the abbreviation of CESC, KICH, in Table S1.
Mention the reference for GSE72096 and GSE115978 datasets in Fig. S2.
Provide the abbreviation of CNA in Fig. S10.
Author Response
We feel great thanks for your professional review work on our article. As you are concerned, there are several problems that need to be addressed. According to your nice suggestions, we have made some corrections to our previous draft.
Point-to-point response:
- It is interesting to know that ACAP1 is a lymphoid lineage specific biomarker and not a myeloid lineage as per your results described in section 3.5. However, in Section 3.3 the results describe prolonged survival in LAML which is a myeloid leukemia. Please explain why is it so?
Response: This is an interesting question. As suggested, we added such discussion in the Discussion section (lines 543-548). We hope that these additional discussions and references can partly explain this question.
- In section 3.5 it is mentioned that ACAP1 expression is negatively correlated with M2 Macrophages. Please explain which data point signifies this in figure7.
Response: Thanks for pointing out this error. We have deleted the "M2 Macrophages" in the revision.
- In discussion section, it is wrong to say ACAP1 studies focused mainly on their role in endocytic recycling function. Please include a description of reports describing ACAP1 as a prognostic biomarker of tumor immunity in recent years including the report on CTNNB1 as a potential biomarker for immunotherapy prognosis in patients with hepatocellular carcinoma.
Response: Thank you for pointing this out. We have mentioned a recent report describing ACAP1 as a biomarker of tumor immunity in ovarian cancer (ref 14, DOI: 10.1089/dna.2020.5596) (lines 89-91). However, CENTB1, but not CTNNB1, is an alias of ACAP1.
- Please discuss if ACAP1`s biomarker roles are dependent on its known functions of endocytic recycling and cellular migration for immunotherapy outcomes in the discussion section.
Response: Thank you for your suggestion. We have discussed this in lines 579-594 in the revision.
- In Fig. S11 pertaining to the results of GSEA analysis, the genes involved in different pathways is not clear, so please add a caption describing main pathways associated.
Response: Thank you for your suggestion. We have added it in method section (lines 195-197 ) and table S2 in the revision.
- Expand ACAP1 and CENTB1 at their first mention in the manuscript.
Response: Thank you for your suggestion. We have added them in lines 81-82 and Table S1 in the revision.
- Expand TIDE for general audience.
Response: Thank you for your suggestion. We have added it in lines 471-472 and Table S1 in the revision.
- Please provide legend for table1 and briefly describe the performance criteria. Also, what is the significance of highlighting some rows in the table? The relevance of number of positive and negative cases is not clear, please describe this in the text.
Response: Thank you for your suggestion. As suggested, we added note to table 1 (lines 859-860) and performance criteria (lines 478-479) in the revision. There is no significance of highlighting in table 1, and we have deleted the highlighting in the revision. We also added the description of “pos” and “neg” in note of table 1 (line 859).
- Indicate the reference numbers for references mentioned in table 1 so it becomes easy for readers to find them.
Response: Thank you for your suggestion. We have added it in Method section (lines 148-152) in the revision.
- Please insert space between author names and year in the table and elsewhere in the manuscript.
Response: Thank you for your suggestion. We have modified them in the revision.
- There is no reference available for following mentioned in table 1. Please revise and provide appropriate references.
Nathanson2017
Liu2019.
Lauss2017
Hugo2016
Ruppin2021
Kim2018
Braun2020
Zhao2019
Uppaluri2020
Response: Thanks for your careful checking. We have added them in Method section (lines 148-152) in the revision.
- Correct the statement in the legend of figure2: Notations in green represent decreased expression in tumor whereas notations in red represent increased gene expression in tumor.
Response: Thanks for your careful checking, and sorry for the wrong descriptions. It has been corrected in the revision.
- Be consistent with the notation of HuH-7 and SK-HEP-1 cells.
Response: Thanks for your careful checking. It has been corrected in the revision.
- Provide appropriate reference of xena-browser used for identification of CpG islands on ACAP1 promoters.
Response: Thank you for your suggestion. We have added it in the revision (line 121, ref 19).
- Line 115, 434- reference Ruppin2021 is not included in the list of references. Reference #28 is wrongly mentioned as Ruppin 2021.
Response: Thanks for your careful checking. We have added it in the revision (line 139), and corrected it.
- Line 114- Reference 26 is not Snyder 2014. It is absent in the reference list.
Response: Thanks for your careful checking. We have corrected it in the revision.
- Line 292 – paracancerous is one word. Correct it.
Response: Thanks for your careful checking. We have corrected it in the revision (line 328).
- 5 heading should read – ACAP1 expression is positively associated with TILs and is essential for... , please correct accordingly.
Response: Thank you for your suggestion. We have corrected it in the revision (lines 416-417).
- Line 438 – Reference Uppaluri 2000 is missing in references.
Response: Thanks for your careful checking. We have added it in the revision (lines 151-152).
- Line 440, 441, 444 – References – Snyder 2014, Lauss 2017 are missing in reference list.
Response: Thanks for your careful checking. We have added it in the revision (line 138 and 149).
- Correct the abbreviation of CESC, KICH, in Table S1.
Response: Thanks for your careful checking. We have corrected it in table s1.
- Mention the reference for GSE72096 and GSE115978 datasets in Fig. S2.
Response: Thanks for your careful checking. We have corrected it in line 131.
- Provide the abbreviation of CNA in Fig. S10.
Response: Thanks for your careful checking. We have added it in the revision.
Reviewer 3 Report
This manuscript by Yi Q. et al. attempts to explore the genetic alterations, expression patterns, prognostic significance, and transcriptional regulation of ACAP1 across a variety of cancer cell types, in addition to the relationship between ACAP1 expression in tumor-infiltrating lymphocytes (TILs) and the response rates to immune checkpoint blockade therapy. The authors identify ACAP1 as a lymphocyte-specific gene and also as an indispensable for the function of lymphocytes under the normal conditions and present data demonstrating that depletion of ACAP1 correlates with the cold immune phenotype and resistance to immune checkpoint blockade therapy in solid tumors. Most of the data in this manuscript are highly novel and informative. Overall, the Reviewer found the manuscript to be clearly and well written, technically solid, and interesting. The authors could significantly strength their manuscript by addressing the following concerns.
Major Comments
1. In Figure 3, ACAP1 was significantly decreased in tissues from most cancer types compared with normal tissue, while significantly increased in those from six cancer types. Could the authors have any thoughts about what the cause of this difference might be? The authors should describe a difference among cancer types in their Discussion.
2. In Figure 4, the authors assessed the prognostic value of the ACAP1 mRNA expression levels for a variety of tumor types. However, as the authors described in their Results, the prognostic significance of ACAP1 expression varies by tumor types, although lower expression of ACAP1 was associated with poor clinical outcomes in the majority of tumor types. This difference merits some comment in the Discussion.
3. Page 2, Line 66–67: The authors should provide more detailed explanation, such as the mechanism by which ACAP1 influences the infiltration process of several immune cells in the ovarian cancer regarding the paper by Zhang J. et al. you cited.
Author Response
We feel great thanks for your professional review work on our article. As you are concerned, there are several problems that need to be addressed. According to your nice suggestions, we have made some corrections to our previous draft.
Point-to-point response:
- In Figure 3, ACAP1 was significantly decreased in tissues from most cancer types compared with normal tissue, while significantly increased in those from six cancer types. Could the authors have any thoughts about what the cause of this difference might be? The authors should describe a difference among cancer types in their Discussion.
Response: We think this is an excellent suggestion. As suggested, we added such discussion in the Discussion section (lines 525-533).
- In Figure 4, the authors assessed the prognostic value of the ACAP1 mRNA expression levels for a variety of tumor types. However, as the authors described in their Results, the prognostic significance of ACAP1 expression varies by tumor types, although lower expression of ACAP1 was associated with poor clinical outcomes in the majority of tumor types. This difference merits some comment in the Discussion.
Response: We think this is an excellent suggestion. As suggested, we added such discussion in the Discussion section (lines 534-557).
- Page 2, Line 66–67: The authors should provide more detailed explanation, such as the mechanism by which ACAP1 influences the infiltration process of several immune cells in the ovarian cancer regarding the paper by Zhang J. et al. you cited.
Response: Thank you for your suggestion. As suggested, we added some explanation in the revision (lines 90-91).
Round 2
Reviewer 1 Report
The authors have addressed all my questions.
Author Response
Thank you again for your valuable suggestion to improve the quality of our manuscript.
Reviewer 2 Report
Authors have not addressed question 1 and Question 4 from previous revision.
Please address these questions appropriately.
Author Response
Thank you again for your valuable suggestion to improve the quality of our manuscript.
Point-to-point response:
Question1 : It is interesting to know that ACAP1 is a lymphoid lineage specific biomarker and not a myeloid lineage as per your results described in section 3.5. However, in Section 3.3 the results describe prolonged survival in LAML which is a myeloid leukemia. Please explain why is it so?
Response: This is a very interesting question, but difficult to accurately address it. We regret that we did not give a satisfactory reply to this question in the previous version of the manuscript. In this version of the manuscript, we try our best to explain it.
“We showed that ACAP1 is a lymphoid lineage specific biomarker, and high ACAP1 level correlates with poor overall survival in LAML. This indicates that high lymphocyte count may be an indicator of poor overall survival in LAML. In 1991, Edward et al. found that positive lymphocyte surface markers (CD2 and CD19) were associated with a more favorable prognosis in AML patients [72]. And two other studies showed that high ALC (absolute lymphocytic count) was associated with prolonged overall survival of AML patients [73,74]. However, a recent study showed that high ALC was associated with shorter remission, decreased relapse-free and overall survival in AML patients who responded to induction therapy [75]. In addition, AML patients with higher ALC showed a lower response rate to induction therapy [73]. There were low frequency of NK cells and high frequency of inhibitory T regulatory cells in the high ALC group [75], which may result in weaker immune responses against residual leukemia remaining after induction therapy, and thus, might explain the poorer outcome of the high ALC group in AML patients after induction therapy. Another possible reason is that lymphocytes in AML are usually exhausted [76-79].”
We have added these discussions in the discussion section of the manuscript. We hope that these additional discussions could partly explain this question.
Question 4: Please discuss if ACAP1`s biomarker roles are dependent on its known functions of endocytic recycling and cellular migration for immunotherapy outcomes in the discussion section.
Response: Thank you for your suggestion. We have added some discussions in the discussion section of the revised manuscript.
For functions of endocytic recycling:
“Previous studies showed that ACAP1 acts as an adaptor for endocytic recycling of intracellular cargos, such as transferrin receptor and integrin β1 (CD29) [8-11]. Cell surface transferrin receptor is required for B and T cell activation, and regulates cell-mediated immune responses [82,83]. Integrin β1 is a representative adhesion molecule for cell-cell and cell-extracellular matrix interactions, and it plays a crucial role in lymphocytes. For example, Integrin β1 on T cells could induce FAK-mediated signaling and subsequent CD40L expression, and then enhance T cell activation [84,85]. Integrin β1 also marked CD4+ and CD8+ T cells maintaining the cytotoxic phenotype [86,87]. It is required for optimal proliferation and cytotoxicity function of NK cells [88,89]. Moreover, integrin β1 expression correlates with the expression of markers for tumor-infiltrating lymphocytes and better long-term survival of patients with melanoma, suggesting a role in anti-tumor response [90]. Therefore, we speculate that ACAP1 is also necessary for the activation and function of these cells, and loss of ACAP1 in lymphocytes would impair their ability of killing tumor cells, leading to poor immunotherapy outcomes.”
For the functions of cellular migration:
“ In addition, it is reported that ACAP1 knockdown inhibits cell migration [9]. Migration of lymphocytes, cytotoxic CD8 T cells in particular, into tumor is essential for its infiltration and for providing efficient defense against tumor cells, thus affecting the outcome of immunotherapy for cancer. Therefore, we speculate that lymphocytes with high expression levels of ACAP1 are more likely to infiltrate into tumors, thus killing tumor cells more effectively. Future studies are required to investigate the possibility.”
Reviewer 3 Report
The authors have responded appropriately to all the comments raised by the Reviewer.
Author Response

(The authors gave the same response as above.)
